# Tractable Bayesian Network Structure Learning with Bounded Vertex Cover Number

**Janne H. Korhonen**
Helsinki Institute for Information Technology HIIT
Department of Computer Science
University of Helsinki
janne.h.korhonen@helsinki.fi

**Pekka Parviainen**
Helsinki Institute for Information Technology HIIT
Department of Computer Science
Aalto University
pekka.parviainen@aalto.fi

## Abstract

Both learning and inference tasks on Bayesian networks are NP-hard in general. Bounded tree-width Bayesian networks have recently received a lot of attention as a way to circumvent this complexity issue; however, while inference on bounded tree-width networks is tractable, the learning problem remains NP-hard even for tree-width 2. In this paper, we propose *bounded vertex cover number Bayesian networks* as an alternative to bounded tree-width networks. In particular, we show that both inference and learning can be done in polynomial time for any fixed vertex cover number bound $k$, in contrast to the general and bounded tree-width cases; on the other hand, we also show that learning problem is W[1]-hard in parameter $k$. Furthermore, we give an alternative way to learn bounded vertex cover number Bayesian networks using integer linear programming (ILP), and show this is feasible in practice.

## 1   Introduction

*Bayesian networks* are probabilistic graphical models representing joint probability distributions of random variables. They can be used as a model in a variety of prediction tasks, as they enable computing the conditional probabilities of a set of random variables given another set of random variables; this is called the *inference* task. However, to use a Bayesian network as a model for inference, one must first obtain the network. Typically, this is done by estimating the network based on observed data; this is called the *learning* task.

Both the inference and learning tasks are NP-hard in general [3, 4, 6]. One approach to deal with this issue has been to investigate special cases where these problems would be tractable. That is, the basic idea is to select models from a restricted class of Bayesian networks that have structural properties enabling fast learning or inference; this way, the computational complexity will not be an issue, though possibly at the cost of accuracy if the true distribution is far from the model family. Most notably, it is known that the inference task can be solved in polynomial time if the network has *bounded tree-width*, or more precisely, the inference task is *fixed-parameter tractable* in the tree-width of the network. Moreover, this is in a sense optimal, as bounded tree-width is necessary for polynomial-time inference unless the exponential time hypothesis (ETH) fails [17].

The possibility of tractable inference has motivated several recent studies also on learning bounded tree-width Bayesian networks [2, 12, 16, 19, 22]. However, unlike in the case of inference, learning a Bayesian network of bounded tree-width is NP-hard for any fixed tree-width bound at least 2 [16]. Furthermore, it is known that learning many relatively simple classes such as paths [18] and polytrees [9] is also NP-hard. Indeed, so far the only class of Bayesian networks for which a polynomial time learning algorithm is known are trees, i.e., graphs with tree-width 1 [5] – it appears that our knowledge about structure classes allowing tractable learning is quite limited.

## 1.1 Structure Learning with Bounded Vertex Cover Number

In this work, we propose *bounded vertex cover number Bayesian networks* as an alternative to the tree-width paradigm. Roughly speaking, we consider Bayesian networks where all pairwise dependencies – i.e., edges in the moralised graph – are *covered* by having at least one node from the vertex cover incident to each of them; see Section 2 for technical details. Like bounded tree-width Bayesian networks, this is a *parameterised* class, allowing a trade-off between the complexity of models and the size of the space of possible models by varying the parameter $k$.

**Results: complexity of learning bounded vertex cover networks.** Crucially, we show that learning an optimal Bayesian network structure with vertex cover number at most $k$ can be done in polynomial time for any fixed $k$. Moreover, vertex cover number provides an upper bound for tree-width, implying that inference is also tractable; thus, we identify a rare example of a class of Bayesian networks where both learning and inference are tractable.

Specifically, our main theoretical result shows that an optimal Bayesian network structure with vertex cover number at most $k$ can be found in time $4^k n^{2k+O(1)}$ (Theorem 5). However, while the running time of our algorithm is polynomial with respect to the number of nodes, the degree of the polynomial depends on $k$. We show that this is in a sense best we can hope for; that is, we show that there is no *fixed-parameter* algorithm with running time $f(k) \operatorname{poly}(n)$ for any function $f$ even when the maximum allowed parent set size is restricted to 2, unless the commonly accepted complexity assumption FPT $\neq$ W[1] fails (Theorem 6).

**Results: ILP formulation and learning in practice.** While we prove that the learning bounded vertex cover Bayesian network structures can be done in polynomial time, the unavoidable dependence on $k$ in the degree the polynomial makes the algorithm of our main theorem infeasible for practical usage when the vertex cover number $k$ increases. Therefore, we investigate using an *integer linear programming* (ILP) formulation as an alternative way to find optimal bounded vertex cover Bayesian networks in practice (Section 4). Although the running time of an ILP is exponential in the worst case, the actual running time in many practical scenarios is significantly lower; indeed, most of the state-of-the-art algorithms for exact learning of Bayesian networks in general [1, 8] and with bounded tree-width [19, 22] are based on ILPs. Our experiments show that bounded vertex cover number Bayesian networks can, indeed, be learned fast in practice using ILP (Section 5).

## 2 Preliminaries

**Directed graphs.** A directed graph $D = (N, A)$ consists of a *node set* $N$ and *arc set* $A \subseteq N \times N$; for a fixed node set, we usually identify a directed graph with its arc set $A$. A directed graph is called a *directed acyclic graph* or *DAG* if it contains no directed cycles. We write $n = |N|$ and $uv$ for arc $(u, v) \in A$. For $u, v \in N$ with $uv \in A$, we say that $u$ is a *parent* of $v$ and $v$ is a *child* of $u$. We write $A_v$ for the *parent set* of $v$, that is, $A_v = \{u \in N : uv \in A\}$.

**Bayesian network structure learning.** We consider the Bayesian network structure learning using the score-based approach [7, 14], where the input consists of the node set $N$ and the *local scores* $f_v(S)$ for each node $v \in N$ and $S \subseteq N \setminus \{v\}$. The task is to find a DAG $A$ – the *network structure* – that maximises the score $f(A) = \sum_{v \in N} f_v(A_v)$.

We assume that the scores $f_v$ are computed beforehand, and that we can access each entry $f_v(S)$ in constant time. We generally consider a setting where only parent sets belonging to specified sets $\mathcal{F}_v \subseteq 2^N$ are permitted. Typically, $\mathcal{F}_v$ consists of parent sets of size at most $k$, in which case we assume that the scores $f_v(S)$ are given only for $|S| \leq k$; that is, the size of the input is $O\left(n\binom{n}{k}\right)$.

**Moralised graphs.** For a DAG $A$, the *moralised graph* of $A$ is an undirected graph $M_A = (N, E_A)$, where $E_A$ is obtained by adding (1) an undirected edge $\{u, v\}$ to $E_A$ for each arc $uv \in A$, and (2) by adding an undirected edge $\{u, v\}$ to $E_A$ if $u$ and $v$ have a common child, that is, $\{uw, vw\} \subseteq A$ in $A$ for some $w \in A$. The edges added to $E_A$ due to rule (2) are called *moral* edges.

**Tree-width and vertex cover number.** A *tree-decomposition* of a graph $G = (V, E)$ is a pair $(\mathcal{X}, T)$, where $T$ is a tree with node set $\{1, 2, \ldots, m\}$ and $\mathcal{X} = \{X_1, X_2, \ldots, X_m\}$ is a collection of subsets of $V$ with $\bigcup_{i=1}^{m} X_i = V$ such that

(a) for each $\{u, v\} \in E$ there is $i$ with $u, v \in X_i$, and

(b) for each $v \in V$ the graph $T[\{i \colon v \in X_i\}]$ is connected.

The *width* of a tree-decomposition $(T, \mathcal{X})$ is $\max_i |X_i| - 1$. The *tree-width* $\mathrm{tw}(G)$ of a graph $G$ is the minimum width of a tree-decomposition of $G$. For a DAG $A$, we define the tree-width $\mathrm{tw}(A)$ as the tree-width of the moralised graph $M_A$ [12].

For a graph $G = (V, E)$, a set $C \subseteq V$ is a *vertex cover* if each edge is incident to at least one vertex in $C$. The *vertex cover number* of a graph $\tau(G)$ is the size of the smallest vertex cover in $G$. As with tree-width, we define the vertex cover number $\tau(A)$ of a DAG $A$ as $\tau(M_A)$.

**Lemma 1.** *For a DAG $A$, we have $\mathrm{tw}(A) \leq \tau(A)$.*

*Proof.* By definition, the moralised graph $M_A$ has a vertex cover $C$ of size $\tau(A)$. We can construct a star-shaped tree-decomposition for $M_A$ with a central node $i$ with $X_i = C$ and a leaf $j$ with $X_j = C \cup v$ for every $v \in N \setminus C$. Clearly, this tree-decomposition has width $\tau(A)$; thus, we have $\mathrm{tw}(A) = \mathrm{tw}(M_A) \leq \tau(A)$. $\qquad\square$

**Structure learning with parameters.** Finally, we give a formal definition for the bounded tree-width and bounded vertex cover number Bayesian network structure learning problems. That is, let $p \in \{\tau, \mathrm{tw}\}$; in the *bounded-$p$ Bayesian network structure learning*, we are given a node set $N$, local scores $f_v(S)$ and an integer $k$, and the task is to find a DAG $A$ maximising score $\sum_{v \in N} f_v(A_v)$ subject to $p(A) \leq k$. For both tree-width and vertex cover number, the parameter $k$ also bounds the maximum parent set size, so we will assume that the local scores $f_v(S)$ are given only if $|S| \leq k$.

## 3 Complexity Results

### 3.1 Polynomial-time Algorithm

We start by making a few simple observations about the structure of bounded vertex cover number Bayesian networks. In the following, we slightly abuse the terminology and say that $N_1 \subseteq N$ is a vertex cover for a DAG $A$ if $N_1$ is a vertex cover of $M_A$.

**Lemma 2.** *Let $N_1 \subseteq N$ be a set of size $k$, and let $A$ be a DAG on $N$. Set $N_1$ is a vertex cover for $A$ if and only if*

(a) *for each node $v \notin N_1$, we have $A_v \subseteq N_1$, and*

(b) *each node $v \in N_1$ has at most one parent outside $N_1$.*

*Proof.* ($\Rightarrow$) For (a), we have that if there were nodes $u, v \notin N_1$ such that $u$ is the child of $v$, the moralised graph $M_A$ would have edge $\{u, v\}$ that is not covered by $N_1$. Likewise for (b), we have that if a node $u \in N_1$ had parents $v, w \notin N_1$, then $M_A$ would have edge $\{v, w\}$ not covered by $N_1$. Thus, both (a) and (b) have to hold if $A$ has vertex cover $N_1$.

($\Leftarrow$) Since (a) holds, all directed edges in $A$ have one endpoint in $N_1$, and thus the corresponding undirected edges in $M_A$ are covered by $N_1$. Moreover, by (a) and (b), no node has two parents outside $N_1$, so all moral edges in $M_A$ also have at least one endpoint in $N_1$. $\qquad\square$

Lemma 2 allows us to partition a DAG with vertex cover number $k$ into a *core* that covers at most $2k$ nodes that are either in a fixed vertex cover or are parents of those nodes (*core nodes*), and a *periphery*

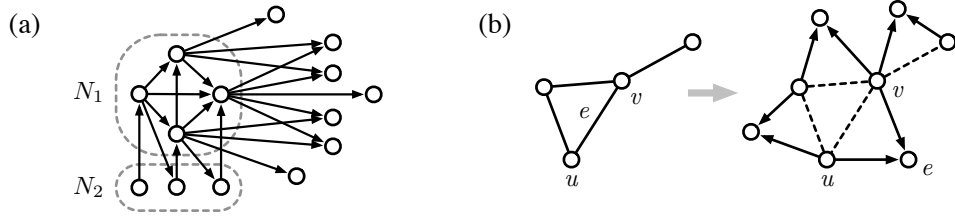

Figure 1: (a) Example of a DAG with vertex cover number 4, with sets $N_1$ and $N_2$ as in Lemma 3. (b) Reduction used in Theorem 6; each edge in the original graph is replaced by a possible v-structure.

containing arcs going into nodes that have no children and all parents in the vertex cover (*peripheral nodes*). This is illustrated in Figure 1(a), and the following lemma formalises the observation.

**Lemma 3.** *Let $A$ be a DAG on $N$ with vertex cover $N_1$ of size $k$. Then there is a set $N_2 \subseteq N \setminus N_1$ of size at most $k$ and arc sets $B$ and $C$ such that $A = B \cup C$ and*

(a) *$B$ is a DAG on $N_1 \cup N_2$ with vertex cover $N_1$, and*

(b) *$C$ contains only arcs $uv$ with $u \in N_1$ and $v \notin N_1 \cup N_2$.*

*Proof.* First, let $N_2 = \left( \bigcup_{v \in N_1} A_v \setminus N_1 \right)$. By Lemma 2, each $v \in N_1$ can have at most one parent outside $N_1$, so we have $|N_2| \le |N_1| \le k$.

Now let $B = \{uv \in A \colon u, v \in N_1 \cup N_2\}$ and $C = A \setminus B$. To see that (a) holds for this choice of $B$, we observe that the edge set of the moralised graph $M_B$ is a subset of the edges in $M_A$, and thus $N_1$ covers all edges of $M_B$. For (b), the choice of $N_2$ and Lemma 2 ensure that nodes in $N \setminus (N_1 \cup N_2)$ have no children, and again by Lemma 2 their parents are all in $N_1$. $\square$

Dually, if we fix the core and peripheral node sets, we can construct a DAG with bounded vertex cover number by the selecting the core independently from the parents of the peripheral nodes. Formally:

**Lemma 4.** *Let $N_1, N_2 \subseteq N$ be disjoint. Let $B$ be a DAG on $N_1 \cup N_2$ with vertex cover $N_1$, and let $C$ be a DAG on $N$ such that $C$ only contains arcs $uv$ with $u \in N_1$ and $v \notin N_1 \cup N_2$. Then*

(a) *$A = B \cup C$ is a DAG on $N$ with vertex cover $N_1$, and*

(b) *the score of $A$ is $f(A) = \sum_{v \in N_1 \cup N_2} f_v(B_v) + \sum_{v \notin N_1 \cup N_2} f_v(C_v)$.*

*Proof.* To see that (a) holds, we observe that $B$ is acyclic by assumption, and addition of arcs from $C$ cannot create cycles as there are no outgoing arcs from nodes in $N \setminus (N_1 \cup N_2)$. Moreover, for $v \in N_1 \cup N_2$, there are no arcs ending at $v$ in $C$, and likewise for $v \notin N_1 \cup N_2$, there are no arcs ending at $v$ in $B$. Thus, we have $A_v = B_v$ if $v \in N_1 \cup N_2$ and $A_v = C_v$ otherwise. This implies that since conditions of Lemma 2 hold for both $B$ and $C$, they also hold for $A$, and thus $N_1$ is a vertex cover for $A$. Finally, the preceding observation implies also that $f_v(A_v) = f_v(B_v)$ for $v \in N_1 \cup N_2$ and $f_v(A_v) = f_v(C_v)$ otherwise, which implies (b). $\square$

Lemmas 3 and 4 give the basis of our strategy for finding an optimal Bayesian network structure with vertex cover number at most $k$. That is, we iterate over all possible $\binom{n}{k}\binom{n-k}{k} = O(n^{2k})$ choices for sets $N_1$ and $N_2$; for each choice, we construct the optimal core and periphery as follows, keeping track of the best found DAG $A^*$:

**Step 1.** To find the optimal core $B$, we construct a Bayesian network structure learning instance on $N_1 \cup N_2$ by removing nodes outside $N_1 \cup N_2$ and restricting the possible choices of parent sets so that $\mathcal{F}_v = 2^{N_1}$ for all $v \in N_2$, and $\mathcal{F}_v = \{S \subseteq N_1 \cup N_2 \colon |S \cap N_2| \le 1\}$ for $v \in N_1$. By Lemma 2, any solution for this instance is a DAG with vertex cover $N_1$. Moreover, this instance has $2k$ nodes, so it can be solved in time $O(k^2 2^{2k})$ using the dynamic programming algorithm of Silander and Myllymäki [23].

**Step 2.** To construct the periphery $C$, we compute the value $\hat{f}_v(N_1) = \max_{S \subseteq N_1} f_v(S)$ and select corresponding best parent set choice $C_v$ for each $v \notin N_1 \cup N_2$; this can be done in time in $O(nk2^k)$ time using the dynamic programming algorithm of Ott and Miyano [21].

**Step 3.** We check if $f(B \cup C) > f(A^*)$, and replace $A^*$ with $B \cup C$ if this holds.

By Lemma 4(a), all DAGs considered by the algorithm are valid solutions for Bayesian network structure learning with bounded vertex cover number, and by Lemma 4(b), we can find the optimal solution for fixed $N_1$ and $N_2$ by optimising the choice of the core and the periphery separately. Moreover, by Lemma 3 each bounded vertex cover DAG is included in the search space, so we are guaranteed to find the optimal one. Thus, we have proven our main theorem:

**Theorem 5.** *Bounded vertex cover number Bayesian network structure learning can be solved in time $4^k n^{2k+O(1)}$.*

### 3.2 Lower Bound

Although the algorithm presented in the previous section runs in polynomial time in $n$, the degree of the polynomial depends on the size of vertex cover $k$, which poses a serious barrier to practical use when $k$ grows. Moreover, the algorithm is essentially optimal in the general case, as the input has size $\Omega\left(n\binom{n}{k}\right)$ when parent sets of size at most $k$ are allowed. However, in practice one often assumes that a node can have at most, say, 2 or 3 parents. Thus, it makes sense to consider settings where the input is restricted, by e.g. considering instances where parent set size is bounded from above by some constant $w$ while allowing vertex cover number $k$ to be higher. In this case, we might hope to do better, as the input size is not a restricting factor.

Unfortunately, we show that it is not possible to obtain a algorithm where the degree of the polynomial does not depend on $k$ even when the maximum parent set size is limited to 2, that is, there is no algorithm with running time $g(k) \operatorname{poly}(n)$ for any function $g$, unless the widely believed complexity assumption FPT $\neq$ W[1] fails. Specifically, we show that Bayesian network structure learning with bounded vertex cover number is W[1]-hard when restricted to instances with parent set size 2, implying the above claim. For full technical details on complexity classes FPT and W[1] and the related theory, we refer the reader to standard texts on the topic [11, 13, 20]; for our result, it suffices to note that the assumption FPT $\neq$ W[1] implies that finding a $k$-clique from a graph cannot be done in time $g(k) \operatorname{poly}(n)$ for any function $g$.

**Theorem 6.** *Bayesian network structure learning with bounded vertex cover number is W[1]-hard in parameter $k$, even when restricted to instances with maximum parent set size 2.*

*Proof.* We prove the result by a parameter-preserving reduction from clique, which is known to be W[1]-hard [10]. We use the same reduction strategy as Korhonen and Parviainen [16] use in proving that the bounded tree-width version of the problem is NP-hard. That is, given an instance $(G = (V, E), k)$ of clique, we construct a new instance of bounded vertex cover number Bayesian network structure learning as follows. The node set of the instance is $N = V \cup E$. The parent scores are defined by setting $f_e(\{u, v\}) = 1$ for each $e = \{u, v\} \in E$, and $f_v(S) = 0$ for all other $v$ and $S$; see Figure 1(b). Finally, the vertex cover size is required to be at most $k$. Clearly, the new instance can be constructed in polynomial time.

It now suffices to show that the original graph $G$ has a clique of size $k$ if and only if the optimal DAG $N$ with vertex cover number at most $k$ has score $\binom{k}{2}$:

($\Rightarrow$) Assume $G$ has a $k$-clique $C \subseteq V$. Let $A$ be a DAG on $N$ obtained by setting $A_e = \{u, v\}$ for each $e = \{u, v\} \subseteq C$, and $A_v = \emptyset$ for all other nodes $v \in N$. All edges in the moralised graph $M_A$ are now clearly covered by $C$. Furthermore, since $C$ is a clique in $G$, there are $\binom{k}{2}$ nodes with a non-empty parent set, giving $f(A) = \binom{k}{2}$.

($\Leftarrow$) Assume now that there is a DAG $A$ on $N$ with vertex cover number $k$ and a score $f(A) \geq \binom{k}{2}$. There must be at least $\binom{k}{2}$ nodes $e = \{u, v\} \in E$ such that $A_e = \{u, v\}$, as these are the only nodes that can contribute to a positive score. Each of these triangles $T_e = \{e, u, v\}$ for $e = \{u, v\}$ must contain at least two nodes from a minimum vertex cover $C$; without loss of generality, we may assume that these nodes are $u$ and $v$, as $e$ cannot cover any other edges. However, this means that $C \subseteq V$ and there are at least $\binom{k}{2}$ edges $e \subseteq C$, implying that $C$ must be a $k$-clique in $G$. $\qquad\square$

# 4 Integer Linear Programming

To complement the combinatorial algorithm of Section 3.1, we will formulate the bounded vertex cover number Bayesian network structure learning problem as an integer linear program (ILP). Without loss of generality, we may assume that nodes are labeled with integers $[n]$.

As a basis for the formulation, let $z_{Sv}$ be a binary variable that takes value 1 when $S$ is the parent set of $v$ and 0 otherwise. The objective function for the ILP is

$$\max \sum_{v \in N} \sum_{S \in \mathcal{F}_v} f_v(S) z_{Sv}.$$

To ensure that the variables $z_{Sv}$ encode a valid DAG, we use the standard constraints introduced by Jaakkola et al. [15] and Cussens [8]:

$$\sum_{S \in \mathcal{F}_v} z_{Sv} = 1 \qquad \forall v \in N \tag{1}$$

$$\sum_{v \in W} \sum_{\substack{S \in \mathcal{F}_v \\ S \cap W = \emptyset}} z_{Sv} \geq 1 \qquad \forall W \subseteq N \colon |W| \geq 1 \tag{2}$$

$$z_{Sv} \in \{0,1\} \quad \forall v \in N, S \in \mathcal{F}_v. \tag{3}$$

Now it remains to bound the vertex cover number of the moralised graph. We introduce two sets of binary variables. The variable $y_{uv}$ takes value 1 if there is an edge between nodes $u$ and $v$ in the moralised graph and 0 otherwise. The variable $c_u$ takes value 1 if the node $u$ is a part of the vertex cover and 0 otherwise. By combining a construction of the moralised graph and a well-known formulation for vertex cover, we get the following:

$$\sum_{S \in \mathcal{F}_v \colon u \in S} z_{Sv} + \sum_{T \in \mathcal{F}_u \colon v \in T} z_{Tu} - y_{uv} \leq 0 \qquad \forall u, v \in N \colon u < v \tag{4}$$

$$z_{Sv} - y_{uw} \leq 0 \qquad \forall v \in N, S \in \mathcal{F}_v \colon u, w \in S, u < w \tag{5}$$

$$y_{uv} - c_u - c_v \leq 0 \qquad \forall u, v \in N \colon u < v \tag{6}$$

$$\sum_{u \in N} c_u \leq k \tag{7}$$

$$y_{uv}, c_u \in \{0,1\} \quad \forall u, v \in N. \tag{8}$$

The constraints (4) and (5) guarantee that $y$-variables encode the moral graph. The constraint (6) guarantees that if there is an edge between $u$ and $v$ in the moral graph then either $u$ or $v$ is included in the vertex cover. Finally, the constraint (7) bounds the size of the vertex cover.

# 5 Experiments

We implemented both the combinatorial algorithm of Section 3.1 and the ILP formulation of Section 4 to benchmark the practical performance of the algorithms and test how good approximations bounded vertex cover DAGs provide. The combinatorial algorithm was implemented in Matlab and is available online[1]. The ILPs were implemented using CPLEX Python API and solved using CPLEX 12. The implementation is available as a part of TWILP software[2].

**Combinatorial algorithm.**  As the worst- and best-case running time of the combinatorial algorithm are the same, we tested it with synthetic data sets varying the number of nodes $n$ and the vertex cover bound $k$, limiting each run to at most 24 hours. The results are shown in Figure 2. With reasonable vertex cover number bounds the polynomial-time algorithm scales only up to about 15 nodes; this is mainly due to the fact that, while the running time is polynomial in $n$, the degree of the polynomial depends on $k$ and when $k$ grows, the algorithm becomes quickly infeasible.

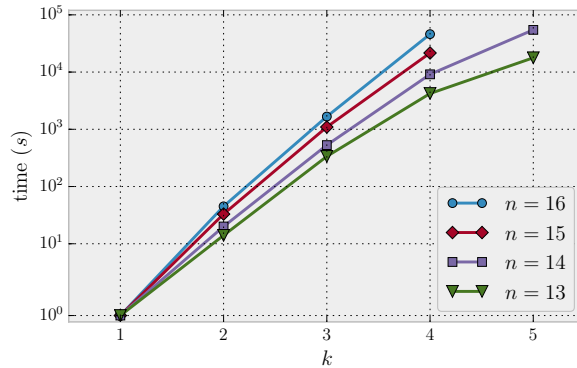

Figure 2: Running times of the polynomial time algorithm. Number of nodes varies from 13 to 16 and the vertex cover number from 1 to 5. For $n = 15$ and $n = 16$ with $k = 5$, the algorithm did not finish in 24 hours.

**Integer linear program.** We ran our experiments using a union of the data sets used by Berg et al. [2] and those provided at GOBNILP homepage[3]. We benchmarked the results against other ILP-based algorithms, namely GOBNILP [8] for learning Bayesian networks without any restrictions to the structure and TWILP [22] for learning bounded tree-width Bayesian networks. In our tests, each algorithm was given 4 hours of CPU time. Figure 3 shows results for selected data sets. Due to space reasons, full results are reported in the supplement.

The results show that optimal DAGs with moderate vertex cover number (7 for flag, 6 for carpo10000) tend to have higher scores than optimal trees. This suggests that often one can trade speed for accuracy by moving from trees to bounded vertex cover number DAGs. We also note that bounded vertex cover number DAGs are usually learned quickly, typically at least two orders-of-magnitude faster than bounded tree-width DAGs. However, bounded tree-width DAGs are a less constrained class, and thus in multiple cases the best found bounded tree-width DAG has better score than the corresponding bounded vertex cover number DAG even when the bounded tree-width DAG is not proven to be optimal. This seems to be the case also if we have mismatching bound, say, 5 for tree-width and 10 for vertex cover number.

Finally, we notice that ILP solves easily problem instances with, say, 60 nodes and vertex cover bound 8; see the results for carpo10000 data set. Thus, in practice ILP scales up to significantly larger data sets and vertex cover number bounds than the combinatorial algorithm of Section 3.1. Presumably, this is due to the fact that ILP solvers tend to use heuristics that can quickly prune out provably non-optimal parts of choices for the vertex cover, while the combinatorial algorithm considers them all.

## 6 Discussion

We have shown that bounded vertex cover number Bayesian networks both allow tractable inference and can be learned in polynomial time. The obvious point of comparison is the class of trees, which has the same properties. Structurally these two classes are quite different. In particular, neither is a subclass of the other – DAGs with vertex cover number $k > 1$ can contain dense substructures, while a path of $n$ nodes (which is also a tree) has a vertex cover number $\lfloor n/2 \rfloor = \Omega(n)$.

In contrast with trees, bounded vertex cover number Bayesian networks have a densely connected "core" , and each node outside the core is either connected to the core or it has no connections. Thus, we would expect them to perform better than trees when the "real" network has a few dense areas and only few connections between nodes outside these areas. On the other hand, bounding the vertex cover number bounds the total size of the core area, which can be problematic especially in large networks when some parts of the network are not represented in the minimum vertex cover.

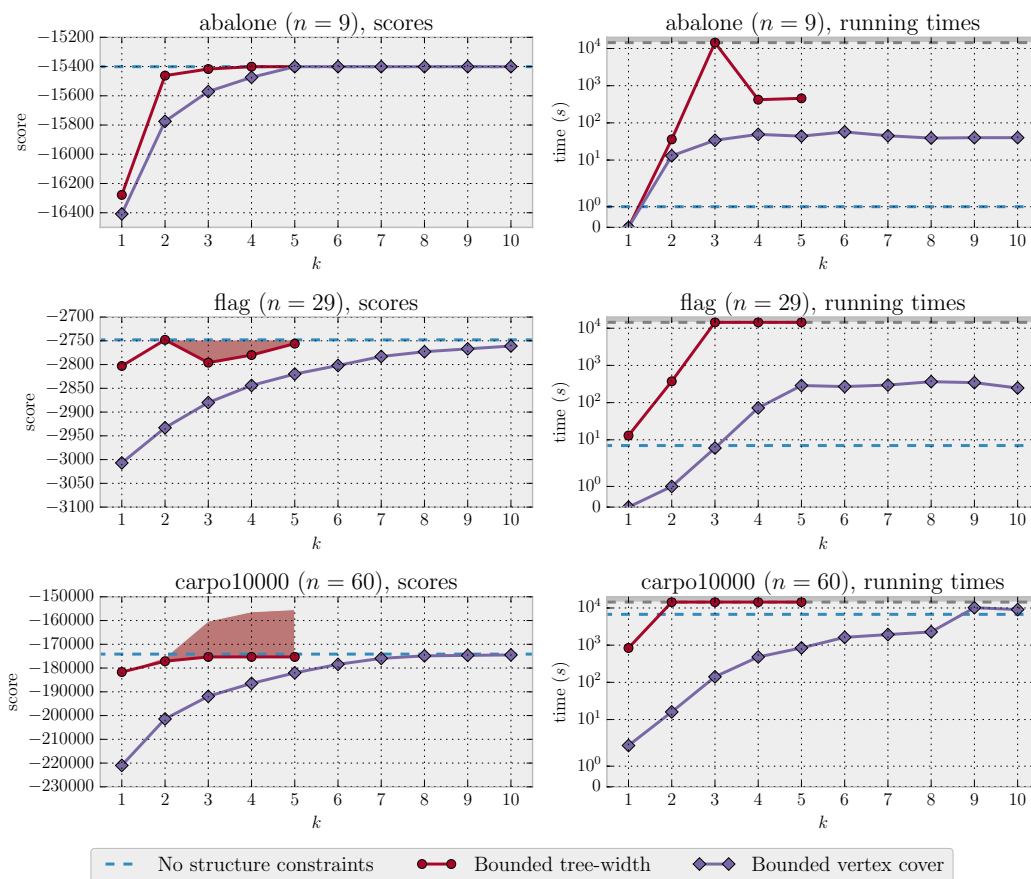

Figure 3: Results for selected data sets. We report the score for the optimal DAG without structure constraints, and for the optimal DAGs with bounded tree-width and bounded vertex cover when the bound $k$ changes, as well as the running time required for finding the optimal DAG in each case. If the computations were not finished at the time limit of 4 hours, we show the score of the best DAG found so far; the shaded area represents the unexplored part of the search space, that is, the upper bound of the shaded area is the best score upper bound proven by the ILP solver.

We also note that bounded vertex cover Bayesian networks have a close connection to naive Bayes classifiers. That is, variables outside a vertex cover are conditionally independent of each other given the vertex cover. Thus, we can replace the vertex cover by a single variable whose states are a Cartesian product of the states of the vertex cover variables; this star-shaped network can then be viewed as a naive Bayes classifier.

Finally, we note some open question related to our current work. From a theoretical perspective, we would like to classify different graph parameters in terms of complexity of learning. Ideally, we would want to have a graph parameter that has a fixed-parameter learning algorithm when we bound the maximum parent set size, circumventing the barrier of Theorem 6. From a practical perspective, there is clearly room for improvement in efficiency of our ILP-based learning algorithm; for instance, GOBNILP uses various optimisations beyond the basic ILP encoding to speed up the search.

### Acknowledgments

We thank James Cussens for fruitful discussions. This research was partially funded by the Academy of Finland (Finnish Centre of Excellence in Computational Inference Research COIN, 251170). The experiments were performed using computing resources within the Aalto University School of Science "Science-IT" project.

## Footnotes

[1]http://research.cs.aalto.fi/pml/software/VCDP/

[2]http://bitbucket.org/twilp/twilp

[3]http://www.cs.york.ac.uk/aig/sw/gobnilp/

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
