[Supplementary Material]

# Supplement for the paper"Tractable Bayesian Network Structure Learning with Bounded Vertex Cover Number"

Janne H. Korhonen and Pekka Parviainen

October 23, 2015

| Data | n | Unbounded | | tree-width = 1 | | tree-width = 2 | | tree-width = 3 | | tree-width = 4 | | tree-width = 5 | |
|---|---|---|---|---|---|---|---|---|---|---|---|---|---|
| | | Score | Time | Score | Time | Score | Time | Score | Time | Score | Time | Score | Time |
| Mildew10000 | 35 | -407644 | 1 | -424311 | 34 | -407821(0.32%) | >14400 | -407644 | 1446 | -407644 | 789 | -407644 | 1723 |
| Mildew1000 | 35 | -47102 | 0 | -47245 | 21 | -47102 | 34 | -47102 | 384 | -47102 | 645 | -47102 | 1104 |
| Mildew100 | 35 | -5720 | 2 | -6000 | 1 | -5935 | 2 | -5786(1.15%) | >14400 | -5746(0.46%) | >14400 | -5720 | 2985 |
| Water10000 | 32 | -128706 | 26 | -129804 | 26 | -129103(0.26%) | >14400 | -129016(0.3%) | >14400 | -128803(0.13%) | >14400 | -128709(0.06%) | >14400 |
| Water1000 | 32 | -13262 | 31 | -13307 | 43 | -13276(0.13%) | >14400 | -13264(0.04%) | >14400 | -13262 | 3102 | -13262 | 4365 |
| Water100 | 32 | -1501 | 28 | -1508 | 17 | -1502(0.03%) | >14400 | -1501(0.06%) | >14400 | -1501 | 5578 | -1501 | 7870 |
| alarm10000 | 37 | -105227 | 786 | -118383 | 67 | -110217(5.53%) | >14400 | -106349(2.59%) | >14400 | -106797(3.07%) | >14400 | -106975(3.34%) | >14400 |
| alarm1000 | 37 | -11240 | 7 | -12315 | 69 | -11774(4.93%) | >14400 | -11469(2.48%) | >14400 | -11291(0.95%) | >14400 | -11273(0.94%) | >14400 |
| alarm100 | 37 | -1349 | 2 | -1473 | 70 | -1443(5.57%) | >14400 | -1455(7.84%) | >14400 | -1412(4.79%) | >14400 | -1401(4.24%) | >14400 |
| asia10000 | 8 | -22466 | 1 | -23037 | 1 | -22469(0.14%) | >14400 | -22466 | 541 | -22466 | 510 | -22466 | 31 |
| asia1000 | 8 | -2317 | 0 | -2372 | 0 | -2329(0.55%) | >14400 | -2317 | 159 | -2317 | 111 | -2317 | 6 |
| asia100 | 8 | -246 | 0 | -246 | 0 | -246 | 2 | -246 | 21 | -246 | 42 | -246 | 2 |
| carpo10000 | 60 | -174131 | 6739 | -181686 | 839 | -177127(0.89%) | >14400 | -175324(8.63%) | >14400 | -175312(10.84%) | >14400 | -175324(11.37%) | >14400 |
| carpo1000 | 60 | -17719 | 106 | -18413 | 1142 | -18192(2.29%) | >14400 | -18436(10.82%) | >14400 | -18271(10.42%) | >14400 | -19112(17.27%) | >14400 |
| carpo100 | 60 | -1829 | 900 | -1974 | 465 | -2050(10.71%) | >14400 | -2065(13.95%) | >14400 | -2101(21.17%) | >14400 | -2131(23.02%) | >14400 |
| hailfinder10000 | 56 | -497632 | 86 | -512293 | 557 | -500054(0.7%) | >14400 | -499777(9.44%) | >14400 | -500313(11.75%) | >14400 | -500313(16.56%) | >14400 |
| hailfinder1000 | 56 | -52473 | 16 | -53401 | 617 | -52478(0.19%) | >14400 | -52593(7.37%) | >14400 | -52616(10.98%) | >14400 | -52605(12.36%) | >14400 |
| hailfinder100 | 56 | -6019 | 0 | -6076 | 41 | -6021 | 182 | -6019 | 7926 | -6034(5.87%) | >14400 | -6041(7.25%) | >14400 |
| insurance10000 | 27 | -132969 | 23 | -144844 | 24 | -142329(6.77%) | >14400 | -138442(4.35%) | >14400 | -133866(1.07%) | >14400 | -133108(0.51%) | >14400 |
| insurance1000 | 27 | -13887 | 1 | -14702 | 15 | -14261(2.63%) | >14400 | -14170(2.04%) | >14400 | -13887 | 785 | -13887 | 1314 |
| insurance100 | 27 | -1686 | 0 | -1728 | 10 | -1711(1.42%) | >14400 | -1692(0.35%) | >14400 | -1686 | 341 | -1686 | 492 |
| kreditfamily | 18 | -16696 | 0 | -16698 | 0 | -16696 | 1 | -16696 | 32 | -16696 | 33 | -16696 | 26 |
| Abalone | 9 | -15401 | 1 | -16278 | 0 | -15462 | 36 | -15417(0.1%) | >14400 | -15401 | 420 | -15401 | 457 |
| adult15N | 15 | -351151 | 2 | -356324 | 2 | -354806(0.81%) | >14400 | -354656(0.94%) | >14400 | -351712(0.16%) | >14400 | -351151 | 283 |
| Flag | 29 | -2748 | 7 | -2803 | 13 | -2748 | 374 | -2796(1.7%) | >14400 | -2780(1.15%) | >14400 | -2756(0.29%) | >14400 |
| Heart | 23 | -2397 | 1 | -2445 | 8 | -2397 | 86 | -2401(0.16%) | >14400 | -2397 | 239 | -2397 | 413 |
| Hepatitis | 20 | -1323 | 1 | -1338 | 12 | -1323 | 45 | -1330(0.6%) | >14400 | -1323 | 232 | -1323 | 184 |
| Horse | 28 | -4525 | 3 | -4579 | 11 | -4561(0.81%) | >14400 | -4525 | 781 | -4525 | 844 | -4525 | 1153 |
| Housing | 14 | -3080 | 33 | -3479 | 2 | -3424(5.01%) | >14400 | -3251(3.29%) | >14400 | -3247(5.34%) | >14400 | -3150(3.44%) | >14400 |
| Voting | 17 | -4643 | 1 | -4658 | 5 | -4643 | 30 | -4643 | 174 | -4643 | 193 | -4643 | 163 |
| Wine | 14 | -1271 | 2 | -1285 | 2 | -1271 | 34 | -1271 | 134 | -1271 | 249 | -1271 | 127 |
| Zoo | 17 | -848 | 0 | -858 | 2 | -848 | 15 | -848 | 70 | -848 | 118 | -848 | 101 |

Table 1: Results for unbounded tree-width and tree-widths 1–5. For each case, the score of an optimal DAG and the running time is reported; if the computations were not finished at the time limit, we report the score of the best DAG found and the gap (the gap is a ratio $|s_1 - s_2|/|s_1|$, where $s_1$ is the score of the best feasible solution and $s_2$ is the smallest known upper bound for the score of an optimal network). Scores and running times are rounded to the nearest integer.

| Data | n | Unbounded | | VC number = 1 | | VC number = 2 | | VC number = 3 | | VC number = 4 | | VC number = 5 | |
|---|---|---|---|---|---|---|---|---|---|---|---|---|---|
| | | Score | Time | Score | Time | Score | Time | Score | Time | Score | Time | Score | Time |
| Mildew10000 | 35 | -407644 | 1 | -509428 | 0 | -473918 | 3 | -454469 | 3 | -446114 | 13 | -439382 | 11 |
| Mildew1000 | 35 | -47102 | 0 | -54412 | 0 | -51411 | 1 | -49772 | 1 | -49202 | 10 | -48759 | 4 |
| Mildew100 | 35 | -5720 | 2 | -6309 | 0 | -6151 | 0 | -5983 | 74 | -5892 | 57 | -5833 | 70 |
| Water10000 | 32 | -128706 | 26 | -160387 | 0 | -154095 | 1 | -148486 | 7 | -142990 | 45 | -138879 | 53 |
| Water1000 | 32 | -13262 | 31 | -16339 | 0 | -15754 | 1 | -15186 | 1 | -14643 | 1 | -14215 | 1 |
| Water100 | 32 | -1501 | 28 | -1791 | 0 | -1732 | 0 | -1682 | 2 | -1633 | 1 | -1588 | 1 |
| alarm10000 | 37 | -105227 | 786 | -172927 | 1 | -154994 | 5 | -137775 | 38 | -130289 | 86 | -124761 | 625 |
| alarm1000 | 37 | -11240 | 7 | -17884 | 1 | -16072 | 2 | -14402 | 6 | -13744 | 22 | -13264 | 86 |
| alarm100 | 37 | -1349 | 2 | -2037 | 0 | -1883 | 2 | -1747 | 2 | -1648 | 6 | -1578 | 10 |
| asia10000 | 8 | -22466 | 1 | -25794 | 0 | -22820 | 2 | -22469 | 10 | -22466 | 21 | -22466 | 20 |
| asia1000 | 8 | -2317 | 0 | -2621 | 0 | -2357 | 0 | -2320 | 3 | -2317 | 6 | -2317 | 8 |
| asia100 | 8 | -246 | 0 | -271 | 0 | -247 | 0 | -246 | 1 | -246 | 1 | -246 | 1 |
| carpo10000 | 60 | -174131 | 6739 | -220979 | 2 | -201424 | 16 | -191893 | 141 | -186470 | 481 | -182037 | 843 |
| carpo1000 | 60 | -17719 | 106 | -22109 | 1 | -20244 | 8 | -19397 | 19 | -18913 | 24 | -18546 | 34 |
| carpo100 | 60 | -1829 | 900 | -2420 | 4 | -2213 | 7 | -2104 | 34 | -2055 | 151 | -2005 | 310 |
| hailfinder10000 | 56 | -497632 | 86 | -604656 | 4 | -586718 | 14 | -571259 | 41 | -558578 | 73 | -547059 | 141 |
| hailfinder1000 | 56 | -52473 | 16 | -62208 | 1 | -60701 | 5 | -59224 | 7 | -57975 | 9 | -56897 | 14 |
| hailfinder100 | 56 | -6019 | 0 | -6788 | 0 | -6664 | 1 | -6543 | 2 | -6437 | 2 | -6339 | 1 |
| insurance10000 | 27 | -132969 | 23 | -190711 | 0 | -172051 | 3 | -162410 | 27 | -154919 | 104 | -147054 | 29 |
| insurance1000 | 27 | -13887 | 1 | -19193 | 0 | -17671 | 1 | -16830 | 12 | -15997 | 7 | -15239 | 9 |
| insurance100 | 27 | -1686 | 0 | -2102 | 0 | -1988 | 5 | -1913 | 3 | -1843 | 10 | -1793 | 5 |
| kreditfamily | 18 | -16696 | 0 | -17213 | 0 | -16978 | 0 | -16821 | 0 | -16762 | 0 | -16725 | 1 |
| Abalone | 9 | -15401 | 1 | -16408 | 0 | -15775 | 13 | -15571 | 34 | -15474 | 49 | -15401 | 44 |
| adult15N | 15 | -351151 | 2 | -396233 | 0 | -369029 | 4 | -357371 | 15 | -355010 | 79 | -353333 | 154 |
| Flag | 29 | -2748 | 7 | -3007 | 0 | -2933 | 1 | -2880 | 6 | -2844 | 72 | -2820 | 288 |
| Heart | 23 | -2397 | 1 | -2867 | 0 | -2746 | 1 | -2631 | 4 | -2531 | 4 | -2476 | 8 |
| Hepatitis | 20 | -1323 | 1 | -1414 | 0 | -1374 | 9 | -1352 | 16 | -1339 | 18 | -1331 | 21 |
| Horse | 28 | -4525 | 3 | -4872 | 0 | -4763 | 0 | -4674 | 6 | -4628 | 22 | -4589 | 7 |
| Housing | 14 | -3080 | 33 | -3715 | 0 | -3494 | 10 | -3400 | 433 | -3292 | 680 | -3198 | 819 |
| Voting | 17 | -4643 | 1 | -4885 | 0 | -4703 | 13 | -4670 | 16 | -4658 | 21 | -4653 | 30 |
| Wine | 14 | -1271 | 2 | -1313 | 0 | -1289 | 23 | -1276 | 35 | -1272 | 27 | -1271 | 25 |
| Zoo | 17 | -848 | 0 | -971 | 0 | -927 | 9 | -890 | 6 | -863 | 5 | -850 | 4 |

Table 2: Results for unbounded vertex cover number and vertex cover numbers 1–5. For each case, the score of an optimal DAG and the running time is reported; if the computations were not finished at the time limit, we report the score of the best DAG found and the gap (the gap is a ratio $|s_1 - s_2|/|s_1|$, where $s_1$ is the score of the best feasible solution and $s_2$ is the smallest known upper bound for the score of an optimal network). Scores and running times are rounded to the nearest integer.

| Data | n | VC number = 6 | | VC number = 7 | | VC number = 8 | | VC number = 9 | | VC number = 10 | |
|---|---|---|---|---|---|---|---|---|---|---|---|
| | | Score | Time | Score | Time | Score | Time | Score | Time | Score | Time |
| Mildew10000 | 35 | -434370 | 15 | -429812 | 15 | -425404 | 20 | -421147 | 25 | -416998 | 15 |
| Mildew1000 | 35 | -48349 | 4 | -47945 | 4 | -47580 | 3 | -47449 | 13 | -47321 | 13 |
| Mildew100 | 35 | -5804 | 192 | -5779 | 367 | -5755 | 162 | -5742 | 220 | -5735 | 146 |
| Water10000 | 32 | -135480 | 64 | -132971 | 44 | -131218 | 42 | -130068 | 90 | -129113 | 76 |
| Water1000 | 32 | -13896 | 4 | -13689 | 8 | -13508 | 9 | -13391 | 17 | -13303 | 50 |
| Water100 | 32 | -1548 | 2 | -1535 | 18 | -1523 | 14 | -1516 | 15 | -1509 | 12 |
| alarm10000 | 37 | -119562 | 417 | -115699 | 872 | -113166 | 2012 | -111372 | 13304 | -109791 | 13620 |
| alarm1000 | 37 | -12757 | 37 | -12449 | 180 | -12168 | 106 | -11983 | 123 | -11792 | 159 |
| alarm100 | 37 | -1535 | 33 | -1510 | 104 | -1476 | 37 | -1449 | 42 | -1422 | 83 |
| asia10000 | 8 | -22466 | 34 | -22466 | 44 | -22466 | 41 | -22466 | 42 | -22466 | 47 |
| asia1000 | 8 | -2317 | 5 | -2317 | 6 | -2317 | 6 | -2317 | 6 | -2317 | 6 |
| asia100 | 8 | -246 | 1 | -246 | 1 | -246 | 1 | -246 | 1 | -246 | 1 |
| carpo10000 | 60 | -178440 | 1624 | -175924 | 1925 | -174860 | 2271 | -174658 | 10174 | -174476 | 9054 |
| carpo1000 | 60 | -18167 | 28 | -17933 | 82 | -17804 | 94 | -17780 | 348 | -17756 | 428 |
| carpo100 | 60 | -1969 | 785 | -1933 | 796 | -1913 | 3612 | -1907(0.6%) | >14400 | -1899(0.74%) | >14400 |
| hailfinder10000 | 56 | -539233 | 632 | -531427 | 226 | -524603 | 232 | -519114 | 340 | -514429 | 290 |
| hailfinder1000 | 56 | -56117 | 23 | -55355 | 20 | -54770 | 42 | -54191 | 33 | -53749 | 47 |
| hailfinder100 | 56 | -6278 | 5 | -6218 | 4 | -6160 | 4 | -6125 | 4 | -6096 | 6 |
| insurance10000 | 27 | -142526 | 118 | -139026 | 152 | -136489 | 181 | -135456 | 244 | -134443 | 493 |
| insurance1000 | 27 | -14745 | 16 | -14398 | 19 | -14183 | 21 | -14074 | 35 | -14001 | 55 |
| insurance100 | 27 | -1763 | 8 | -1742 | 12 | -1720 | 9 | -1708 | 11 | -1701 | 19 |
| kreditfamily | 18 | -16699 | 1 | -16696 | 0 | -16696 | 0 | -16696 | 0 | -16696 | 0 |
| Abalone | 9 | -15401 | 57 | -15401 | 45 | -15401 | 39 | -15401 | 40 | -15401 | 40 |
| adult15N | 15 | -351891 | 52 | -351470 | 51 | -351243 | 52 | -351151 | 48 | -351151 | 47 |
| Flag | 29 | -2802 | 270 | -2783 | 296 | -2773 | 365 | -2767 | 344 | -2761 | 247 |
| Heart | 23 | -2452 | 17 | -2428 | 14 | -2418 | 26 | -2409 | 27 | -2403 | 24 |
| Hepatitis | 20 | -1327 | 44 | -1323 | 17 | -1323 | 24 | -1323 | 27 | -1323 | 22 |
| Horse | 28 | -4568 | 14 | -4557 | 50 | -4546 | 31 | -4539 | 52 | -4530 | 35 |
| Housing | 14 | -3135 | 788 | -3111 | 3130 | -3088 | 5622 | -3080 | 2690 | -3080 | 1393 |
| Voting | 17 | -4647 | 36 | -4643 | 30 | -4643 | 21 | -4643 | 33 | -4643 | 22 |
| Wine | 14 | -1271 | 23 | -1271 | 26 | -1271 | 21 | -1271 | 29 | -1271 | 19 |
| Zoo | 17 | -849 | 9 | -848 | 8 | -848 | 9 | -848 | 12 | -848 | 6 |

Table 3: Results for vertex cover numbers 6–10. For each case, the score of an optimal DAG and the running time is reported; if the computations were not finished at the time limit, we report the score of the best DAG found and the gap (the gap is a ratio $|s_1 - s_2|/|s_1|$, where $s_1$ is the score of the best feasible solution and $s_2$ is the smallest known upper bound for the score of an optimal network). Scores and running times are rounded to the nearest integer.