[Reviews · NeurIPS 2015]

Submitted by Assigned_Reviewer_1

The paper describes a new class of Bayes nets for which inference and structure learning can be done in polynomial time.

This is the class of Bayes nets with a bounded vertex cover number.

So far, the only other class of Bayes nets for which inference and structure learning is tractable is the class of trees.

Hence, this is an important contribution that advances our understanding of tractable probabilistic graphical models.

The paper also describes two algorithms to find the best Bayes net structure for a bounded vertex cover k, however it is not clear whether practitioners would want to use those algorithms.

For a fixed k, those algorithms are polynomial in the number of nodes n, but in practice it is not clear what k to use. So one would probably start with a small k and then increase k if there is some time left.

Unfortunately, the algorithms scale exponentially with k, so the search would have to be restricted to small k's. It is not clear either why one would restrict the search to Bayes net with a bounded vertex cover.

I understand that a bounded vertex cover implies that inference will be tractable, but this is also the case for Bayes nets with a bounded tree width. So it seems to me that practitioners would want to do the largest search possible in the space of Bayes nets that allow tractable inference. Since that space is too large for an exhaustive search, then it would make sense to focus on structures that are likely to arise in practice.

It is not clear that Bayes nets with a bounded vertex cover are more likely to occur in practice than other types of tractable Bayes nets with an unbounded vertex cover.

As explained in the paper, Bayes nets with a bounded vertex cover tend to have a star shape with a dense core, but I'm unaware of real problems with such a structure. To verify whether it is better to prioritize the search by increasing the bound on the vertex cover instead of the tree width, could you report two curves (one for prioritizing the search based on the vertex cover number and the other based on the tree-width) that show how the accuracy improves with time?

The paper is clear and well written.

I noticed one minor typo. On the first line of page 3, the width of a tree decomposition should be *max_i* |X_i|-1.

The experiments include many benchmarks, which is great.

However it is difficult to tell what are the trends in those large tables of numbers.

Can you bold the best result for each problem and then indicate how many problems each algorithm ended up winning?

Can you also report the standard error for K-fold cross validation or some statistical test, otherwise it is not clear when some results are really better.
Summary: The paper shows that finding an optimal Bayes net structure with a bounded vertex cover number can be done in polynomial time with respect to the number of nodes.

This is an important contribution that advances our understanding of tractable probabilistic graphical models.

Submitted by Assigned_Reviewer_2

It may be possible/beneficial to alter the constraints (4)-(8) to get tighter linear relaxations. For example, note that

z{b,c}_a + z{a,c}_b + z_{a,b}_c - y_{ab} \leq 0
Summary: Bounded vertex cover (being an upper bound on treewidth) is an attractive constraint for BN learning and so progress in this direction is useful. Providing both a polynomial-time algorithm and a practical ILP implementation is good.

Submitted by Assigned_Reviewer_3

The paper shows that learning an optimal Bayesian network (BN) with a bounded vertex cover can be done in polynomial time. The vertex cover can be unbounded while the treewidth is bounded. This means that networks with a bounded vertex cover are tractable with respect to inference, as well as learning. The polynomial time algorithm is still inefficient due to dependency on the vertex cover. Thus, the paper provides an ILP formulation to learn efficiently in some cases.

The paper is mostly clear. I haven't seen the vertex cover used before as a measure of complexity of BNs. The idea is generally acceptable. Despite the celebrity of the treewidth, alternative measures enrich the theory of Bayesian networks and provide a different perspective on what can be learned in polynomial time.

In my opinion, the result presented is not well motivated. What kind of networks that appear in practice will have a bounded vertex cover?

A network with few variables can be learned in polynomial time? In my opinion, the paper is just proposing a smart variant of the former sentence: a network with a constant number of interacting variables (N1 + N2), along with many others that are always leaf nodes and can be learned efficiently using dynamic programming. Of course the number of variables can be unbounded while the vertex cover is bounded. The applications where that occurs is still not fully investigated in the paper. Without that, it is hard to evaluate the significance of the proposed method.

Does one have to decide which nodes fall in N1 and N2 before learning? Doesn't that constraint the search space of BNs with bounded vertex cover?
Summary: The paper shows that learning an optimal Bayesian network (BN) with a bounded vertex cover can be done in polynomial time. The motivation should be enhanced. The idea of using the vertex cover appears to be a smart variant of a trivial case.

Submitted by Assigned_Reviewer_4

Learning and inference on general Bayesian networks is a hard task. While it is well-known that inference can be efficiently computed on bounded tree width graphs, there is no known parameter that allows efficient learning. This paper presents such a parameter - the vertex cover. It gives a proof that models with a vertex cover k can be learned in O(4^k n^{2k+O(1)) - a polynomial in the number of vertices $n$. Moreover, the authors prove that bounding the number of parents cannot improve this result.

The paper is very clear and easy to read, and gives an important result. However, and as stated by the authors, its practical influence is very limited due to the exponential dependence on k.

Several points: From step 1, one may conclude that bounding the number of possible parents could be a good parameter as vertex cover. Can you elaborate on this? I think that specifying the complexity of the two algorithm used in steps 1 and 2 will be helpful for understanding. In the proof of theorem 5 you used the "without loss of generality" claim - can you explain it? I am not sure it will hold if u or v have only one edge. Please use bold font on the best result in table 1.

Some typos: Line 110 - I believe it should be max not min. Line 246 - you are missing an if.
Summary: it is well-written paper giving a good-to-have result.

Author Feedback
Author rebuttal: We thank reviewers for their comments and suggestions.

GENERAL RESPONSE

We first want to briefly address two points touching comments by several reviewers.

(R1, R2) In our opinion, the motivation for this work is the specific problem of learning networks that allow fast inference. In particular, we want to ensure fast inference also in cases where the true distribution has high complexity e.g. in terms of tree-width; thus, we approximate the true distribution by learning a "simple" model that captures as much of the distribution as possible, while ensuring that inference remains tractable. Thus, this approach can be applied regardless of the structure of the true distribution; approximation of course may not be as good when the true distribution is "complex".

As pointed out by the reviewers, learning bounded tree-width networks would achieve the same objective. However, learning bounded tree-width BNs is NP-hard and seems to be *more* difficult than the learning optimal BN (see e.g. Berg et al. 2014, Parviainen et al. 2014), hence the search for alternative parameters.

(R1, R3, R4) We also want to emphasise a view that the experiments should not be seen as a competitive comparison of the algorithms. In our view, the goal in our tests is to see whether (a) that there are cases where bounded vertex cover BNs give a reasonable approximation of a distribution with high tree-width and (b) bounded vertex cover number BNs can be learned in reasonable time.

Indeed, as all algorithms perform exact optimisation over the specified class of models, the optimal unbounded network always has highest possible score for the given scoring functions. Moreover, the other classes are nested in the sense that DAGs with vertex cover number (tree-width) at most k are a subclass of DAGs with vertex cover number (tree-width) at most k+1; thus, the optimal network of the latter class has always at least as high score as the optimal network of the former. Each problem type is solved by a specialised algorithm: when there are no constraints on the structure of the network we use GOBNILP, when the tree-width is bounded we use TWILP and when the vertex cover number is bounded we use our ILP algorithm (the combinatorial algorithm does not scale up and is not included, but it would provide same scores as our ILP, as both find an optimal network).

We agree that interesting results could be highlighted and visualised better.

R2:

Sets N1 and N2 do not need to be known beforehand; the combinatorial algorithm iterates over all possible choices (lines 196-198) to ensure that all bounded vertex cover number DAGs are included in the search. Indeed, this is the sole reason for the factor n^2k in the running time, as the optimal DAG for fixed N1 and N2 can be found in O(nk2^k + k4^k) time (Steps 1-2 in the main algorithm).

R3:

Bounding the number of possible parents is a necessary but not sufficient condition. Learning BNs with bounded number of parents is NP-hard for any bound greater than 1 (Chickering 1996) and bounding the number of parents does not bound tree-width and thus inference is hard in the worst case. Thus, bounding the number of parents (without any additional assumptions) yields neither tractable learning nor tractable inference.

To elaborate on the part starting with "without loss of generality" in the proof of Theorem 5, we have that the nodes {u, v, e} form a triangle in the moralised graph of solution A; that is, there are undirected edges u-v, v-e and e-u in the moralised graph (i.e., u and v are adjacent to at least two edges both). Since C is assumed to be vertex cover, it must contain at least two of the nodes u,v,e to cover those three edges. The "wlog" part now means that if, say, u and e are in C, we can alter C by replacing e with v to obtain a new vertex cover that does not contain e; the only edge that was covered only by e was e-v, which is now covered by v.

R4:

The discussion on lines 117-120 shows that if a graph G has vertex cover number k then it is possible to construct a tree-decomposition of width k for G using the presented procedure; thus, G has tree-width at most k. The corresponding result for DAGs follows by examining the moralised graph, that is, tw(A) = tw(M_A) <= \tau(M_A) = \tau(A). We will state this observation as a full lemma and elaborate on the proof if possible.